# Preparation of Polyaniline/Emulsion Microsphere Composite for Efficient Adsorption of Organic Dyes

**DOI:** 10.3390/polym12010167

**Published:** 2020-01-09

**Authors:** Yuanli Liu, Liushuo Song, Linlin Du, Peng Gao, Nuo Liang, Si Wu, Tsuyoshi Minami, Limin Zang, Chuanbai Yu, Xu Xu

**Affiliations:** 1Guangxi Key Laboratory of Optical and Electronic Materials and Devices, The Guangxi Key Laboratory of Theory and Technology for Environmental Pollution Control, College of Materials Science and Engineering, Guilin University of Technology, Guilin 541004, China; lyuanli@glut.edu.cn (Y.L.); jiandandexuan@163.com (L.S.); linlindu0916@163.com (L.D.); penggaoglut@163.com (P.G.); nuoliangscun@163.com (N.L.); wusi@mpip-mainz.mpg.de (S.W.); zanglimin0705@163.com (L.Z.); 2Institute of Industrial Science, The University of Tokyo, 4-6-1 Komaba, Meguro-ku, Tokyo 153-8505, Japan; tminami@iis.u-tokyo.ac.jp

**Keywords:** polyaniline, emulsion microsphere, methyl orange, adsorption

## Abstract

Surface-functionalized polymeric microspheres have wide applications in various areas. Herein, monodisperse poly(styrene–methyl methacrylate–acrylic acid) (PSMA) microspheres were prepared via emulsion polymerization. Polyaniline (PANI) was then coated on the PSMA surface via in situ polymerization, and a three-dimensional (3D) structured reticulate PANI/PSMA composite was, thus, obtained. The adsorption performance of the composite for organic dyes under different circumstances and the adsorption mechanism were studied. The obtained PANI/PSMA composite exhibited a high adsorption rate and adsorption capacity, as well as good adsorption selectivity toward methyl orange (MO). The adsorption process followed pseudo-second-order kinetics and the Langmuir isotherm. The maximum adsorption capacity for MO was 147.93 mg/g. After five cycles of adsorption–desorption, the removal rate remained higher than 90%, which indicated that the adsorbent has great recyclability. The adsorbent materials presented herein would be highly valuable for the removal of organic dyes from wastewater.

## 1. Introduction

The wide use of organic dyes in textile, leather, printing, and cosmetics industries resulted in a large amount of colored wastewater, seriously threatening ecological environment and human health [1]. Azo dyes account for about 50% of organic dyes in the world [2,3,4], they are toxic and can accumulate in the human body through the food chain, jeopardizing health and even causing cancers [5]. Methyl orange (MO), as a typical anionic azo dye, is widely used in various industrial fields. However, MO is difficult to remediate from water with chemical and biological methods due to its high chemical stability [6]. Thus, the efficient removal of MO from water is a challenge and worldwide concern.

A number of methods were developed for removal of pollutants from wastewater [7,8,9,10,11], such as adsorption, electrochemical treatment, coagulation/flocculation, ion exchange, photocatalysis, membrane separation, etc. Among them, the adsorption method attracted extensive attention due to its advantages, such as easy operation, low cost, high efficiency, no secondary pollution, and easy industrialization [11]. Although some natural adsorbents and synthetic adsorbents showed a certain adsorption capacity for dyes [12,13,14,15,16,17,18], some drawbacks including low adsorption capacity, poor selectivity, and poor reusability still exist. Therefore, it is urgent to develop a novel adsorbent for the removal of MO from aquatic environments.

Polyaniline (PANI) is widely used in the treatment of dye-contaminated wastewater because of its non-toxicity, low cost, simple preparation, excellent ion exchange performance, and good stability in the environment [19,20]. The interaction between the positively charged PANI and the negatively charged organic dyes leads to PANI having promising potential as an effective adsorbent material. However, pure PANI shows poor adsorption capacity because it undergoes agglomeration during the preparation process. The goal of the present study is to prepare high-performance PANI-based adsorbent materials for organic dyes.

It was reported that the adsorption capacity depends mainly on the number of binding sites (e.g., charged functional groups) [21]. From this point of view, PANI-fabricated three-dimensional (3D) reticulate structures should contain more binding sites compared to linear polymers and, thus, exhibit enhanced adsorption capacity. Some research groups prepared dye adsorbents that contain PANI-based networks [22,23,24]. For example, Aliabadi [22] et al. prepared polypyrrole (PPy), PANI, and their nanocomposites through micro-emulsion polymerization and studied the adsorption performance for anionic dyes. The results showed that the prepared adsorbents had excellent adsorption performance for anionic dyes; however, they showed poor reusability and poor selectivity, thus inevitably resulting in the waste of adsorbents.

In this study, we used styrene (St), methyl methacrylate (MMA), and acrylic acid (AA) as monomers to prepare monodisperse poly(styrene–methyl methacrylate–acrylic acid) (PSMA) emulsion spheres via one-step emulsion polymerization. Subsequently, aniline monomer was coated on the surface of PSMA to prepare polyaniline/core–shell emulsion spheres (PANI/PSMA) composites with a 3D network. The as-prepared PANI/PSMA composite showed good adsorption performance and reusability. The effects of pH, temperature, adsorption time, and dye concentration on the adsorption properties were studied. Moreover, the selectivity and recycling performance of this composite for mixed dyes (anionic and cationic dyes) were also studied.

## 2. Materials and Methods

### 2.1. Materials

Sodium dodecylbenzene sulfonate (SDBS), ammonium bicarbonate (NH_4_HCO_3_), basic alumina (Al_2_O_3_), aniline (ANI), ferric chloride (FeCl_3_), methyl orange (MO), tartrazine (TTZ), amaranth (ART), methylene blue (MB), and methyl violet (MV) with analytical purity were purchased from Aladdin Biochemical Technology Co., Ltd. (Shanghai, China). Styrene (St), methyl methacrylate (MMA), acrylic acid (AA), and ammonium persulfate (APS) with analytical purity were purchased from Sigma-Aldrich (Saint Louis, MI, USA). HCl, NaOH, Na_3_PO_4_, CuSO_4_·5H_2_O, Mg(NO_3_)_2_·6H_2_O, and NaCl (A.R.) were provided by Xilong Chemical Co., Ltd., Shantou, China. The water used in all the experiments was deionized water, which was prepared with a water purification system (AXLM1820). Styrene was purified with a basic alumina chromatographic column before use, ANI was refined by vacuum distillation under N_2_ protection, and other materials were used without further purification.

### 2.2. Preparation of PANI/PSMA

The core–shell PSMA nano-emulsion spheres were prepared following the literature [25]. Firstly, 19.018 g of St, 1.001 g of MMA, 1.001 g of AA, 0.383 g of NH_4_HCO_3_, and 0.003 g of SDBS were dissolved in 100 g of deionized water at 70 °C. After stirring for 30 min, 5.0 g of deionized water containing 0.484 g of APS was dropwise added to the system at 80 °C. After stirring for 10 h, the mixture gave PSMA without additional treatment. Subsequently, 6.0 g of PSMA and 2 g of ANI were dissolved in 50 g of deionized water, followed by ultrasound treatment for 15 min at 35 °C. Then, 50 mL of FeCl_3_ solution (5.481 g of FeCl_3_ dissolved in 50 mL of 0.5 mol/L HCl) was dropwise added to the reaction system. After stirring for 12 h, the product was filtered and washed three times with deionized water and anhydrous ethanol. The sample was dried at 60 °C under vacuum for 8–10 h. Accordingly, PANI/PSMA was obtained. The preparation and adsorption procedures are given in Figure 1.

### 2.3. Adsorption Experiment

The adsorption experiment was performed as follows: PANI/PSMA was dispersed in the dye solution, and the mixture was stirred in a thermostatic oscillator. The pH was adjusted with NaOH (0.1 M) and HCl (0.1 M). A series of dye solutions with different concentrations were prepared, and the absorbance values were firstly measured at an appropriate wavelength for the plot of calibration curves. Then, the absorbance was measured with an ultraviolet–visible light spectrophotometer over a specific time until the adsorption equilibrium was reached. The removal rate for dyes (% removal), adsorption capacity at time t (Q_t_, mg/g), and equilibrium adsorption capacity (Q_e_, mg/g) of PANI/PSMA were calculated with the following formulas:%Removal = (C_0_ − C_e_)/C_0_ × 100,(1)
Q_t_ = (C_0_ − C_t_)/m × V,(2)
Q_e_ = (C_0_ − C_e_)/m × V,(3)
where C_0_ and C_e_ are the initial and equilibrium concentrations (mg/L) of dyes in the solution, C_t_ is the concentration (mg/L) at time t (min), V (L) is the volume of the solution, and m (mg) is the mass of PANI/PSMA adsorbent. 

### 2.4. Selective Adsorption

Typically, 10-mL cationic dye (MB, 0.1 mM) and 10-mL MO (0.1 mM) solutions were mixed in a plastic bottle. Then, 20 mg of PANI/PSMA was added to the above solution. The mixture was oscillated for 6 h. After the adsorption process, the mixture was filtered to determine dye concentration.

### 2.5. Adsorption–Desorption Recycling Experiment

Firstly, 45 mg of adsorbent was added to a 30-mL MO solution (150 mg/L). The mixture was oscillated at 25 °C for 6 h. Then, the mixture was separated with a centrifuge to measure the dye concentration of supernatant. The used PANI/PSMA was immersed in NaOH solution (0.1 M) for desorption for three times; then, it was collected via centrifugation and dried at 60 °C. Finally, the treated adsorbent was used for re-adsorption. Cyclic adsorption–desorption was carried out six times.

### 2.6. Adsorption in Simulated Industrial Wastewater (SIW)

The SIW was prepared according to the “Integrated wastewater discharge standard” (GB8978-1996) and “Discharge standard for water pollutants of textile dyeing and finishing industry” (GB4287-2012). The compositions of SIW include aniline (1 mg/L), Na_3_PO_4_ (12 mg/L), CuSO_4_·5H_2_O (4.3 mg/L), Mg(NO_3_)_2_·6H_2_O (9 mg/L), and NaCl (200 mg/L), and the pH of SIW was 6.0. Then, 50 mg of adsorbent was added to 20 mL of SIW containing MO (10 mg/L). After adsorption of 3 h at 25 °C and pH 6.0, the adsorption effect of MO was measured. Thus, 20 mg of PANI/PSMA was added to 100 mL of MO-contaminated SIW and 100 mL of MO-contaminated aqueous solution (MO concentration: 200 mg/L). After adsorption of 6 h at 25 °C and pH 6.0, the adsorbents were removed, and the adsorption capacities were calculated. The adsorption capacity of prepared adsorbents toward MO in mixed dye-contaminated wastewater was tested as well, and the adsorption effect of PANI/PSMA for MO was determined in 20 mL of SIW containing 10 mg/L MO and 10 mg/L MB. 

### 2.7. Measurements and Characterizations

Fourier-transform infrared (FT-IR) measurements were performed with a Thermo Nexus 470FT-IR spectrometer (Nicolet Company, Waltham, MA, USA). A small amount of sample was ground and pressed with KBr in a mortar. Thermogravimetric analysis (TGA) was performed with a Q500 thermogravimeter (TA Company, Newcastle, DE, USA). The temperature was increased from room temperature to 800 °C at a rate of 20 °C/min. SEM and energy-dispersive X-ray spectroscopy (EDS) were performed with a S-4800 field-emission scanning electron microscope, and the samples were sprayed with gold. Zeta potential was measured with a Zetasizer3000HS nanoparticle-size and zeta potential analyzer (Malvern Company, Malvern, UK). X-ray diffraction (XRD) analysis was performed with an X′Pert PRO wide-angle X-ray scatterer (Panalytical, Almelo, the Netherlands) with a 2θ range of 5°–80°. Ultraviolet (UV) characterization was performed with an ultraviolet–visible light spectrophotometer (PerkinElmer Co., Ltd., Shanghai, China).

## 3. Results

### 3.1. The Structural Characterization of PANI/PSMA

Figure 2a shows the infrared spectra of PANI, PSMA, and PANI/PSMA. For PANI, the peaks at 1567, 1471, 1112, and 796 cm^−1^ were assigned to C=C stretching of the quinone ring, C=C stretching of the benzene ring, N=Q=N (Q = quinone rings stretching, and C–H bending of 1,4-disubstituted aromatic rings, respectively. Moreover, the bands at 1295 and 1236 cm^−1^ were attributed to aromatic conjugated C–N stretching. The results suggested that the PANI was successfully prepared [26]. The characteristic peaks of PSMA were as follows: 2921 cm^−1^, 3025 cm^−1^ (=C–H stretching of benzene ring), 1452 cm^−1^, 1493 cm^−1^ (the vibration of the benzene ring skeleton), 697 cm^−1^, 756 cm^−1^ (the out-of-plane bending vibration of C–H bonds in the benzene ring), 1029 cm^−1^ (the in-plane deformation vibration of H atoms in monosubstituted benzene ring), 2848 cm^−1^ (absorption of methylene), 1731 cm^−1^ (C=O stretching), 1199 cm^−1^ (C–O–C stretching), and 3440 cm^−1^ (–OH stretching). The results confirmed that PSMA was composed of poly(styrene–methyl methacrylate–acrylic acid). In the spectrum of PANI/PSMA, the peaks at 2923 cm^−1^ (=C–H of benzene), 1479 cm^−1^ (the vibration of benzene ring skeleton), 2852 cm^−1^ (the vibration of methylene), and 1569 cm^−1^ (C=C stretching of quinone ring) indicated that PANI and PSMA were successfully combined. Figure 2b shows the TGA curves of PANI, PSMA, and PANI/PSMA. The obvious weight loss of PSMA was observed at 350–450 °C, indicating that PSMA started to decompose at about 350 °C to form small molecules. The organic matter decomposed completely at 450 °C. On the other hand, the weight loss of PANI involved three stages. The three stages at 50–250 °C, 250–458 °C, and 458–800 °C were due to the loss of water, the loss of bound water and the decomposition of PANI into small molecules, and carbonization, respectively [27]. Similarly, the weight of PANI/PSMA also decreased at 50–250 °C due to the loss of water. The PANI/PSMA exhibited a significant weight loss of about 80% at 300–516 °C owing to the decomposition of a large part of PSMA and small part of PANI. When the temperature was higher than 100 °C, slight weight loss occurred thanks to the decomposition of the PANI skeleton. The thermal decomposition temperature of PANI/PSMA (499 °C) increased by 79 °C compared with PSMA (420 °C), which was attributed to strong intermolecular interactions between PANI and PSMA on the surface, thus improving the thermal stability of the PANI/PSMA. The results further proved that PANI was successfully combined with PSMA. Appendix A (Appendix A) shows the XRD patterns of PANI and PANI/PSMA. The PANI exhibited three characteristic peaks at 2θ of 15.08°, 20.81°, and 25.34° corresponding to the 011, 020, and 200 crystal planes, respectively, indicating that PANI was polycrystalline [28,29]. The PANI/PSMA exhibited similar peaks, which suggested that the crystallinity of PANI and PANI/PSMA was similar, i.e., partially crystallized.

Figure 3 shows the SEM images of PSMA, PANI, and the PANI/PSMA composite. In Figure 3a, the microspheres have uniform particle size, regular shape, high sphericity, and good monodispersity (coefficient of variability (CV) = 4.7%). Based on the particle size measurement and Gauss fitting, the mean particle size of PSMA microspheres was determined to be about 317 ± 15 nm. In Figure 3b, PANI was composed of intertwined fibers and a small number of agglomerates. From Figure 3c, it can be seen that PANI/PSMA was covered by reticulate PANI with an enlarged surface area [28]. The N_2_ adsorption–desorption curves (Appendix A, Appendix A) also showed that PANI/PSMA (41.7 m^2^/g) had a larger specific surface area than PANI (22.8 m^2^/g), further demonstrating that PANI/PSMA could provide more effective adsorption sites than PANI.

### 3.2. Adsorption Characteristics of PANI/PSMA for MO

#### 3.2.1. Effects of pH

Because the intermolecular interactions between adsorbents and dye molecules are significantly affected by the pH, pH is a key parameter for dye adsorption [30,31]. The initial pH of solution determines the charges on the surface of the adsorbent. Appendix A (Appendix A) shows the zeta potential profiles of PSMA, PANI, and PANI/PSMA at different pH. The values of zeta potential of PSMA were about −60 in the pH range of 4–10, which indicated that PSMA possessed excellent stability. The pH values of PANI and PANI/PSMA were 6.0 and 4.7 at their isoelectric points, respectively. When the pH was lower than the isoelectric point, the imines and amines in PANI and PANI/PSMA had positive charges. Figure 4a shows the adsorption capacities of PANI and PANI/PSMA for MO at different pH. The results showed that the adsorption capacities decreased gradually with the increase of pH. This was because the basic imino and amino groups of adsorbents produced positive charges in an acidic environment, resulting in improved electrostatic attraction between adsorbents and anionic dye MO. With the increase of pH, the imino and amino groups were deprotonated, causing a smaller number of positive charges. As a result, the electrostatic attraction between adsorbents and MO decreased gradually, and the adsorption capacities decreased accordingly. Moreover, the adsorption capacity of PANI/PSMA was greater than that of PANI, indicating that the combination of PANI and PSMA exhibited a better adsorption performance. The adsorption capacities of PANI and the PANI/PSMA composite were greater at a lower pH.

#### 3.2.2. Effects of the Dye Type

Five representative dyes were selected to study the adsorption behavior toward PANI/PSMA. The initial concentration of these dyes was 100 mg/L. As shown in Figure 4b, the adsorption capacities of PANI/PSMA for anionic dyes (MO, TTZ, and ART) were much higher than those for cationic dyes (MB and MV). The structure of the dyes is shown in Appendix A (Appendix A). For example, the adsorption capacity for MO reached 152.9 mg/g, about 47 times higher than that for cationic MB and MV dyes, respectively. The images of different dyes before and after adsorption by PANI/PSMA are shown in Appendix A (Appendix A). The excellent adsorption performance of PANI/PSMA toward anionic dyes could be attributed to the strong interactions between the positively charged amino groups in PANI/PSMA and negatively charged SO_3_^−^ groups in the anionic dyes. In contrast, MB and MV molecules contain positively charged amino groups, which electrostatically repulse PANI/PSMA, resulting in a poor adsorption performance [32].

#### 3.2.3. Adsorption Kinetics

Adsorption rate is one of the most important indices of adsorptive performance for an adsorbent. With two different initial MO concentrations, the effect of adsorption time on the adsorption of MO is shown in Figure 5a. The adsorption capacity increased with the increase of adsorption time until adsorption equilibrium was reached. Because the adsorption process was slow, the increase of adsorption time was favorable to the interactions between MO molecules and PANI/PSMA adsorbent. The adsorption capacity of PANI/PSMA increased sharply within 60 min and reached an equilibrium state after about 120 min. The high adsorption rate, resulting from the strong electrostatic attraction between the positively charged amines in PANI/PSMA and negatively charged anionic dyes, has obvious advantages in practical applications. For the understanding of the adsorption mechanism, the pseudo-first-order model, pseudo-second-order model, and intraparticle diffusion model were separately fitted with the adsorption data [33,34]. The kinetic parameters were calculated according to the corresponding slope and intercept values in the fitting (Figure 5b–d), as depicted in Table 1.

As illustrated in Figure 5b–d, the correlation coefficients (*R^2^*) of the pseudo-second-order kinetic model were 0.9995 (20 mg/L) and 0.9984 (30 mg/L), higher than those of the pseudo-first-order kinetic model and intraparticle diffusion model. The results show that the adsorption process conformed with the pseudo-second-order kinetic model (Figure 5c). In addition, the Q_e_ values based on the pseudo-second-order kinetic model were 81.30 (20 mg/L) and 116.41 mg/g (30 mg/L), in good agreement with the experimental data (79.37 and 112.10 mg/g, respectively). Because the pseudo-second-order kinetic model is based on the hypothesis of chemical adsorption, the interaction between amino groups in PANI/PSMA and anionic dye molecules was the main driving force for adsorption in the present study.

For a better understanding of rate-determining factors in the adsorption process, the adsorption data were analyzed on the basis of the intraparticle diffusion model [35,36] as described in Table 1. The correlation coefficients (*R^2^*) were 0.8077 (20 mg/L) and 0.9173 (30 mg/L). In the fitting of the intraparticle diffusion model, the fitting curve did not pass through the origin (Figure 5d), indicating that intraparticle diffusion was not the only rate-determining factor. It is supposed that electrostatic attraction, ion exchange, and some other factors may also affect the adsorption rate.

#### 3.2.4. Adsorption Isotherms

For a better understanding of the interactions between dye molecules and the adsorbent surface, the adsorption isotherms were studied as shown in Figure 6. In Figure 6a, the adsorption capacity of PANI/PSMA for MO increased gradually with the increase in initial dye concentration and reached an equilibrium state. When the dye concentration was high, the driving force was strong in the adsorption process, thus overcoming the mass transfer resistance of dye molecules from the aqueous to solid phase and improving the contact between dye molecules and binding sites on the adsorbent. For the study on the maximum adsorption capacity, Langmuir, Freundlich, and Temkin isothermal adsorption models were selected to analyze the adsorption data [37,38,39]. The fitting results and calculated adsorption isotherm constants are shown in Figure 6b–d and Table 2. The correlation coefficient (*R*^2^) of the Langmuir model was 0.9999, higher than that of the other two models. Therefore, the adsorption process agreed well with the Langmuir isothermal adsorption model. After further calculation, the maximum adsorption capacity Q_m_ of PANI/PSMA for MO was found to be 147.93 mg/g, higher than most of the reported PANI-containing adsorption materials (Appendix A, Appendix A), showing that the PANI/PSMA adsorbent has great potential in the adsorption of MO from aqueous solutions.

#### 3.2.5. Adsorption Thermodynamics

Temperature has a significant influence on the adsorption process, and the influence can be reflected by thermodynamic parameters, such as Gibbs free energy change (∆G^0^), enthalpy change (∆H^0^), and entropy change (∆S^0^). After the adsorption process for MO, ∆G^0^ (kJ·mol^−1^), ∆S^0^ (J·mol^−1^·K^−1^), and ∆H^0^ (J·mol^−1^) were calculated according to the van’t Hoff equation [40,41] as follows:ln(Q_e_/C_e_) = ∆S/R − ∆H/RT,(4)
where Q_e_ (mg/g) denotes the amount of dye adsorbed per gram of PANI/PSMA, C_e_ (mg/L) represents the equilibrium concentration of dyes, R stands for the ideal gas constant (8.314 J.mol^−1^·K^−1^), and T (K) denotes the reaction temperature. As shown in Figure 7a, the adsorption capacity of PANI/PSMA for MO increased with the increase of reaction temperature. Figure 7b shows that the correlation coefficient (*R*^2^) between ln(Q_e_/C_e_) and 1/T was 0.9998, which indicates that the adsorption data were in good agreement with the van′t Hoff equation. ∆G^0^ (kJ.mol^−1^) can be calculated according to the following formula:∆G = ∆H − T∆S.(5)

The calculated thermodynamic parameters are shown in Table 3. Both ΔH and ΔS were positive, indicating that the adsorption on PANI/PSMA was endothermic, and that a higher temperature should be more favorable to the adsorption, which is consistent with the experimental results shown in Figure 7a. In addition, the negative ΔG value indicates that the adsorption process was spontaneous.

#### 3.2.6. Selective Adsorption Experiments

In wastewater treatment, adsorption selectivity is a key index of the performance of adsorbents [42,43]. Although many materials can adsorb MO, few materials show adsorption selectivity for MO. Thanks to the numerous positively charged amine groups on the surface of PANI/PSMA, anionic dyes can be exclusively adsorbed on PANI/PSMA from dye mixtures. The selectivity in the adsorption of a mixture of MO/MB on PANI/PSMA was tested, and the results are shown in Figure 8. Figure 8b shows that MB and MO dyes exhibited two intense absorption bands independently before the adsorption. After the addition of PSMA/PANI, the intensity of the absorption band corresponding to anionic dye MO decreased significantly, while that of cationic dye MB did not change significantly, indicating that PANI/PSMA can selectively adsorb anionic dyes from dye mixtures. As shown in Figure 8a, after the adsorption, the color of anionic dye MO faded, and the color of the mixture was close to that of cationic dye MB, which indicated that PANI/PSMA has high selectivity for anionic dyes.

#### 3.2.7. Adsorption–Desorption Recycling Experiments

In practice, the adsorption–desorption recyclability is also a key index of the performance of adsorbents. In the adsorption process, electrostatic attraction was the main reason for the adsorption of MO onto PANI/PSMA. Under alkaline conditions, the zeta potential of PANI/PSMA was negative (Appendix A, Appendix A), indicating weak electrostatic attraction between MO and PANI/PSMA. Therefore, the desorption of MO from the surface of PANI/PSMA can be achieved by treating it with an alkaline solution [25]. As shown in Figure 9, the removal rate of MO remained unchanged in the first four runs. In the fifth run, the removal rate slightly decreased to 91.13%, possibly due to the cleavage of polymer chains caused by the repeated treatment with acid/alkali during the cycles. Thus, PANI/PSMA showed good reusability for the removal of MO.

#### 3.2.8. EDS Analysis before and after Adsorption

The adsorption of MO onto PANI/PSMA was confirmed by energy-dispersive X-ray spectroscopy (Figure 10). Before the adsorption, C and N elements were observed at 0.27 keV in the PANI/PSMA sample, and O was observed at 0.54 keV, indicating that PANI and PSMA were successfully combined. After the adsorption, S element was observed at 2.31 keV, confirming that MO was successfully adsorbed onto the surface of PANI/PSMA.

#### 3.2.9. The Adsorption of SIW

In order to study the adsorption effect of PANI/PSMA adsorbent in industrial wastewater, we studied the adsorption effect in SIW. As can be seen from Figure 11b, the adsorption peak of MO almost completely disappeared in SIW after adsorption by PANI/PSMA. The photographs before and after adsorption also exhibited the same phenomenon (Figure 11a). The adsorption capacity in SIW (140 mg/g) slightly changed compared to that in aqueous solution (150 mg/g). This confirmed that PANI/PSMA was effective for the adsorption of MO in SIW. Figure 11c,d show the photographs and absorption spectra of a mixture dye (MO/MB) in SIW before and after adsorption by PANI/PSMA. After adsorption, the peak of MO decreased significantly, while the peak of MB remained almost unchanged, suggesting that PANI/PSMA had good adsorption selectivity toward MO in SIW. Thus, the PANI/PSMA adsorbent has potential application value in actual industrial wastewater.

## 4. Conclusions

A PANI/PSMA composite was prepared through emulsion polymerization and in situ polymerization. FT-IR, TGA, and SEM results showed that PANI was uniformly coated on the surface of PSMA, and a three-dimensional reticulate structure was successfully fabricated. The as-prepared composite exhibited a fast adsorption rate, high adsorption capacity, excellent adsorption selectivity, and exceptional reusability, and the adsorption process was endothermic and spontaneous, in good agreement with the pseudo-second-order kinetic model and Langmuir isothermal adsorption model. PSMA improved both the dispersion of PANI and the adsorption performance toward MO. Therefore, the PANI/PSMA composite proved to be an excellent adsorbent for dye-contaminated wastewater treatment.

## Figures and Tables

**Figure 1 polymers-12-00167-f001:**
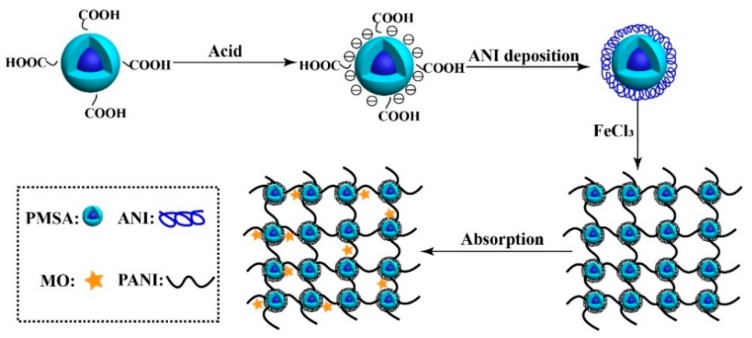
Schematic illustration of the preparation of polyaniline (PANI)/poly(styrene–methyl methacrylate–acrylic acid) (PSMA) adsorbent.

**Figure 2 polymers-12-00167-f002:**
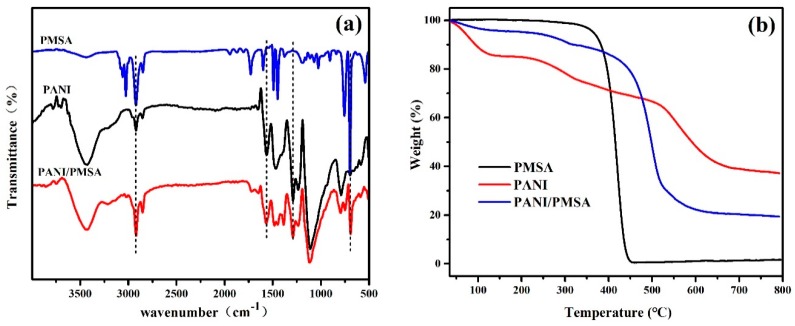
Fourier-transform infrared (FT-IR) curves (**a**) and thermogravimetric analysis (TGA) curves (**b**) for PANI, PSMA, and PANI/PSMA.

**Figure 3 polymers-12-00167-f003:**
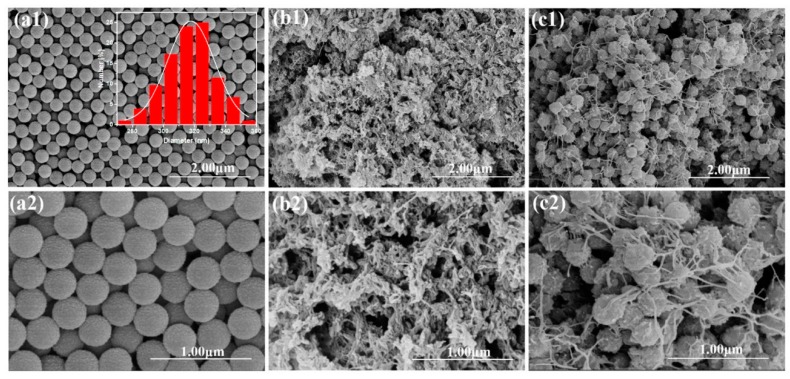
SEM photographs of PSMA (**a1–2**), PANI (**b1–2**), and PANI/PSMA(**c1–2**) (the inset graph in SEM image shows the size distribution measurements of PSMA).

**Figure 4 polymers-12-00167-f004:**
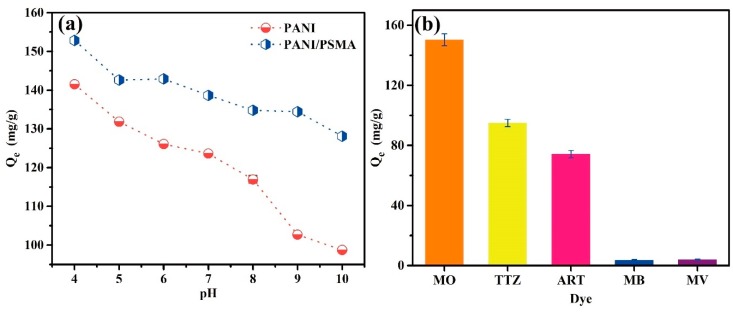
(**a**) The adsorption performance of PANI and PANI/PSMA for methyl orange (MO) at different pH. (**b**) Adsorption capacity of PANI/PSMA adsorption for different dyes.

**Figure 5 polymers-12-00167-f005:**
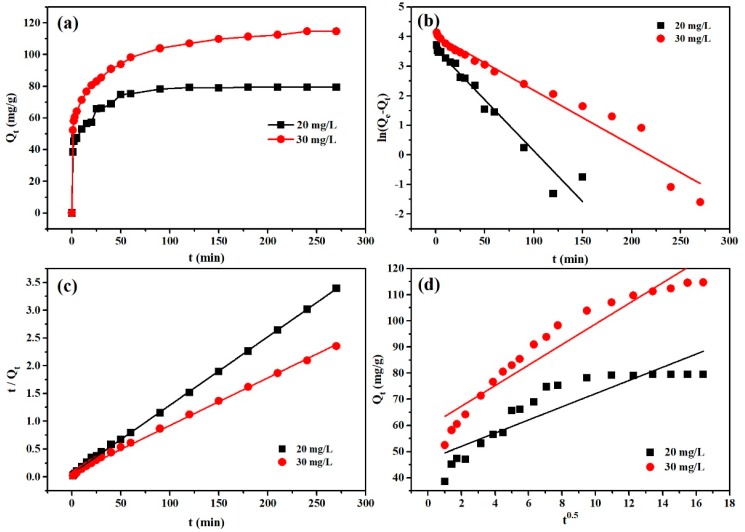
(**a**) Effect of contact time on adsorption of MO by PANI/PSMA at pH 4.0 and 25 °C; (**b**) pseudo-first-order kinetic model plots; (**c**) pseudo-second-order kinetic model plots; (**d**) intraparticle diffusion model plots.

**Figure 6 polymers-12-00167-f006:**
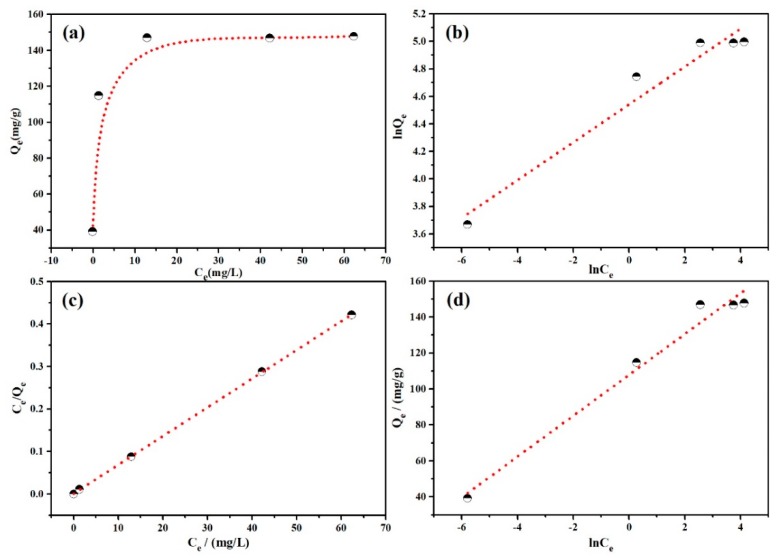
(**a**) Adsorption isotherms for the adsorption of MO by PANI/PSMA at pH 4.0 and 25 °C; (**b**) Freundlich isotherm model plots; (**c**) Langmuir isotherm model plots; (**d**) Temkin model plots.

**Figure 7 polymers-12-00167-f007:**
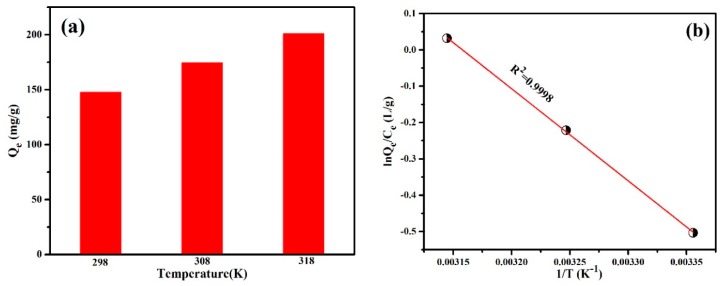
(**a**) Adsorption capacities of PANI/PSMA for MO at different temperature with initial dye concentration of 100 mg/L (pH = 4.0). (**b**) Plots of lnQe/Ce against 1/T for the adsorption of MO onto the PANI/PSMA.

**Figure 8 polymers-12-00167-f008:**
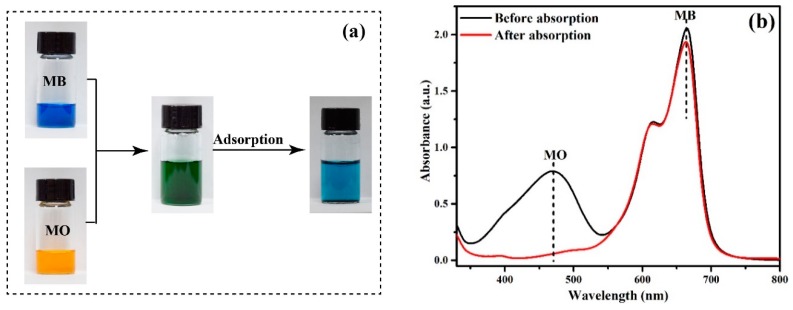
Photographs (**a**) and absorption spectra (**b**) of methylene blue (MB)/MO mixed solutions before and after adsorption by PANI/PSMA for 360 min.

**Figure 9 polymers-12-00167-f009:**
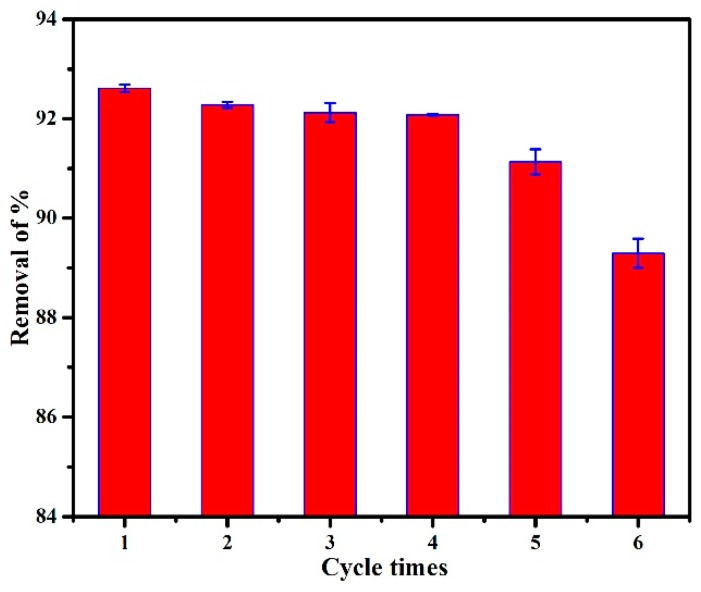
Removal efficiency of the PANI/PSMA in six successive cycles of desorption–adsorption.

**Figure 10 polymers-12-00167-f010:**
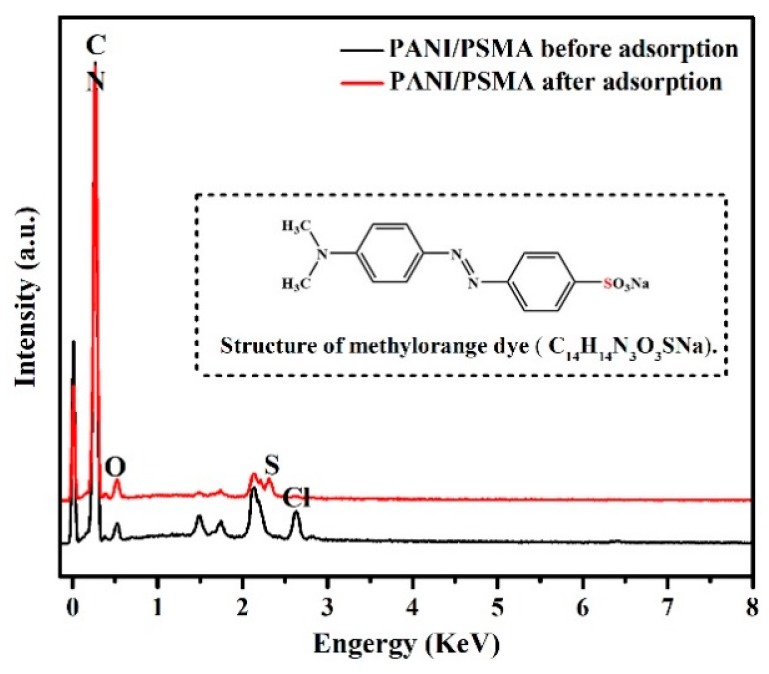
Energy-dispersive X-ray spectroscopy (EDS) spectra of the PANI/PSMA before and after adsorption (the inset in EDS spectra shows the structure of methyl orange dye).

**Figure 11 polymers-12-00167-f011:**
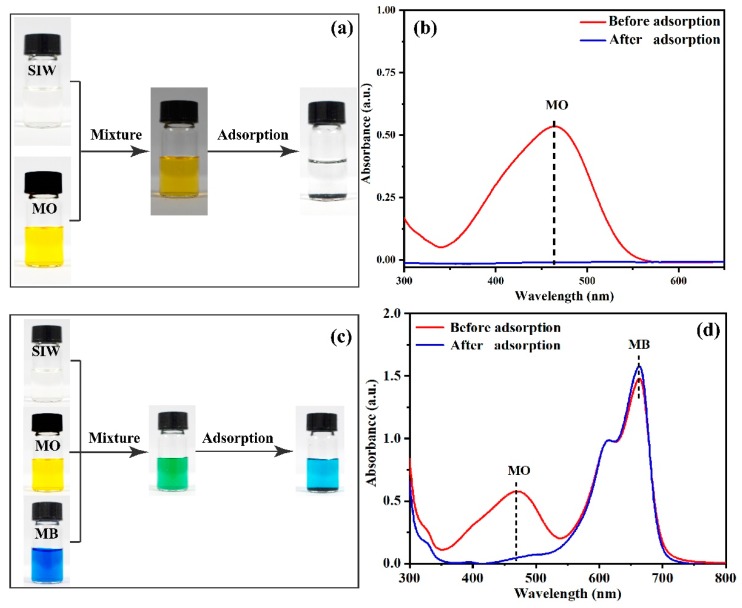
The photographs (**a**) and absorption spectra (**b**) of MO simulated industrial wastewater (SIW), and the photographs (**c**) and absorption spectra (**d**) of MO/MB-containing SIW before and after adsorption by PANI/PSMA.

**Table 1 polymers-12-00167-t001:** Adsorption kinetic parameters of the adsorption of methyl orange (MO) onto the polyaniline (PANI)/poly(styrene–methyl methacrylate–acrylic acid) (PSMA) adsorbent at pH 4.0 and 25 °C. The initial concentrations were 20 and 30 mg/L.

	Q_e-exp_ (mg/g)	Pseudo-Second-Order			Pseudo-First-Order			Intraparticle Diffusion		
		t/Q_t_ = 1/K_2_Q_e_^2^ + t/Q_e_			ln(Q_e_ − Q_t_) = lnQ_e_ − K_1_t			Q_t_ = K_i_t^0.5^ + C		
		Q_e-cal_ (mg/g)	K_2_ (g/mg·min)	*R^2^*	Q_e-cal_ (mg/g)	K_1_ (1/min)	*R^2^*	K_i_	C	*R* ^2^
20	79.37 ± 0.21	81.301	0.00275	0.9995	36.270	0.0345	0.9575	2.522	46.937	0.8077
30	112.10 ± 2.96	116.414	0.00114	0.9984	56.940	0.0185	0.9580	3.938	59.453	0.9173

**Table 2 polymers-12-00167-t002:** Adsorption isotherm parameters for the adsorption of MO onto the PANI/PSMA at pH 4.0 and 25 °C. The contact time was 360 min.

Isotherm Model	Parameters	MO
Langmuir: C_e_/Q_e_ = C_e_/Q_m_ + 1/Q_m_K_L_	Q_m_/(mg·g^−1^)	147.929
	K_L_/(L·mg^−1^)	0.180
	*R* ^2^	0.9999
Freundlich: lnQ_e_ = lnK_F_ + b_F_lnC_e_	K_F_/(mg·g^−1^)	93.785
	b_F_	0.138
	*R* ^2^	0.9544
Temkin: Q_e_ = BlnK_T_ + BlnC_e_	K_T_/(L·mg^−1^)	13266.535
	B/(KJ^−2^·mol^−2^)	11.367
	*R* ^2^	0.9786

**Table 3 polymers-12-00167-t003:** Thermodynamic parameters for the adsorption of MO onto the PANI/PSMA at pH 4.0.

Temperature (K)	∆G^0^ (kJ·mol^−1^)	∆S^0^ (J·mol^−1^·K^−1^)	∆H^0^ (KJ·mol^−1^)
298	−4.996	16.823	0.0168
308	−5.164		
318	−5.333		

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
