# Peer review of "Preparation of Polyaniline/Emulsion Microsphere Composite for Efficient Adsorption of Organic Dyes"

_polymers, 2020, doi:10.3390/polym12010167_

Round 1

Reviewer 1 Report

In the manuscript Authors prepared monodisperse PSMA microspheres via emulsion polymerization and functionalized them PANI. As a result they obtained a 3D PANI/PSMA composite. The PANI/PSMA composite was used as a adsorption material and it was tested with methyl orange (MO).

The manuscript can be published in Polymers after minor revision. My questions are remarks about the manuscript are as follows:

Authors write in 2.1 Materials that "Deionized water was homemade". I think it could be better to describe, even shortly, the method used to obtain deionized water for experiments. 

Please describe method of preparation SEM samples, did Authors used any Au/carbon sputtering on the top of the sample?

Authors write that they obtained microspheres with 'good monodispersity'. Please give the standard deviation (SD) to 317 nm. Having these Authors could calculate CV. If CV if below 5% it can be claimed that monodispersity is good.

At Figure 4a: the line between points should be dotted, as it is a "guide for an eye". The continuous line suggests that there are values between points, but they were measured only for certain pH.

At Figure 4b: can Authors give standard deviations?

Also Figure 9: can Authors give standard deviations?

Reviewer 2 Report

The paper is well preapred and may be accepted for publication after minor revisions: 

100: What was the procedure for determination of the equilibrium adsorption capacity? How was the fact of achieving equilibrium confirmed?

Figure 6: The points spacing is not equal for logarithmical coordinates (i.e. fig 6b) The data between first and second concentration points may improve the fitting.

Reviewer 3 Report

The manuscript  „ID polymers-645187“ focused on a topic relevant not only for materials scientists but also for environmental engineering. The use of polyaniline as dye absorber has however been studied for a long time. This does not mean that the topic is not interesting anymore, but unfortunately, the presented manuscript does not provide any new crucial information. To be more concrete, the initial idea of the study, that adsorption capacity depends mainly on the number of binding sites, is well known and recently a number of strategies (e.g. preparation of composites, nano-tubular or nano-globular forms of conducting polymers etc.) have been developed and tested. Here presented way, the preparation of polymeric microspheres, do not provide any significant novelty or at least this novelty is not evident from the manuscript. The methodology is well arranged in terms of material characterization, but the main topic – adsorption study is based only on the model system, not on for example a set of pollutants presented in environment or wastewater from the industry.  

Although the study is well arranged, the used methods are relevant and results are clearly presented I do not recommend to publish this study as the novelty is low.

Round 2

Reviewer 3 Report

Authors cover one problem of the study and provide a further study on simulated waste water. The novelty of manuscript is however on the same level as before so I do not recommend to publish this study.
